# A substrate-induced gating mechanism is conserved among Gram-positive IgA1 metalloproteases

Jasmina S. Redzic[1], Jeremy Rahkola[2], Norman Tran[3], Todd Holyoak ⬛[3], Eunjeong Lee[1], Antonio Javier Martín-Galiano ⬛[4], Nancy Meyer[5], Hongjin Zheng ⬛[1] & Elan Eisenmesser ⬛[1✉]

The mucosal adaptive immune response is dependent on the production of IgA antibodies and particularly IgA1, yet opportunistic bacteria have evolved mechanisms to specifically block this response by producing IgA1 proteases (IgA1Ps). Our lab was the first to describe the structures of a metal-dependent IgA1P (metallo-IgA1P) produced from Gram-positive Streptococcus pneumoniae both in the absence and presence of its IgA1 substrate through cryo-EM single particle reconstructions. This prior study revealed an active-site gating mechanism reliant on substrate-induced conformational changes to the enzyme that begged the question of whether such a mechanism is conserved among the wider Gram-positive metallo-IgA1P subfamily of virulence factors. Here, we used cryo-EM to characterize the metallo-IgA1P of a more distantly related family member from Gemella haemolysans, an emerging opportunistic pathogen implicated in meningitis, endocarditis, and more recently bacteremia in the elderly. While the substrate-free structures of these two metallo-IgA1Ps exhibit differences in the relative starting positions of the domain responsible for gating substrate, the enzymes have similar domain orientations when bound to IgA1. Together with biochemical studies that indicate these metallo-IgA1Ps have similar binding affinities and activities, these data indicate that metallo-IgA1P binding requires the specific IgA1 substrate to open the enzymes for access to their active site and thus, largely conform to an "induced fit" model.

[1] Department of Biochemistry and Molecular Genetics, School of Medicine, University of Colorado Denver, School of Medicine, Aurora, CO 80045, USA. [2] Mucosal and Vaccine Research Program Colorado, Division of Infectious Disease, University of Colorado Denver School of Medicine and Denver Veterans Affairs Medical Center, Aurora, CO 80045, USA. [3] Department of Biology, University of Waterloo, Waterloo, ON N2L 3G1, Canada. [4] Core Scientific and Technical Units, Instituto de Salud Carlos III (ISCIII), Majadahonda, Spain. [5] Pacific Northwest Cryo-EM Center, Oregon Health and Science University, Portland, OR 97201, USA. ✉email: Elan.Eisenmesser@ucdenver.edu

Many invasive bacteria block the initial mucosal host response during colonization by cleaving host immunoglobulin A1 (IgA1) via a bacterially encoded IgA1 protease (IgA1P). Such bacterial IgA1Ps have evolved to cleave host IgA1 at the hinge region of the heavy chain (HC) (Fig. 1a), which decouples the processes of antigen recognition and the immune response by phagocytic recognition of the IgA1-FC[1,2]. This simultaneously masks the bacteria from any further host immune response, as it leaves the remaining cleaved HC and light chain (LC) that comprise the IgA1-FAB[3,4]. Pathogens that secrete IgA1Ps include Gram-positive bacteria that encode the M26-class of metal-dependent IgA1Ps (metallo-IgA1Ps, also known as ZmpA), which are reliant on a catalytic zinc and include *Streptococcus pneumoniae* (*S. pneumoniae*), *Streptococcus sanguis* (*S. sanguis*), and *Streptococcus oralis* (*S. oralis*), but also Gram-negative bacteria that encode serine-IgA1Ps reliant on a catalytic serine within a trypsin-like domain, including *Haemophilus influenzae* (*H. influenzae*), *Neisseria gonorrhoeae* (*N. gonorrhoeae*), and *Neisseria meningitidis* (*N. meningitidis*)[5–7]. Although the *H. influenzae* serine-IgA1P was structurally elucidated over a decade ago[8], it is only recently that the first structure of a metallo-IgA1P encoded by *S. pneumoniae* was solved by our lab using cryo-EM[9]. This recent *S. pneumoniae* IgA1P study revealed the atomic-resolution details of the first member of the metallo-IgA1P class of enzymes and was the first structure of any IgA1P in complex with its cognate IgA1 substrate. Interestingly, in order to engage the substrate, a ~10 Å "gating mechanism" was necessary to facilitate IgA1 binding. Considering the ubiquitous but sequence-divergent nature of these metallo-IgA1Ps across Gram-positive bacteria, the results of our previous study beg the question of whether this gating mechanism is broadly conserved throughout other Gram-positive metallo-IgA1P virulence factors. To address this, we extended our focus on *S. pneumoniae* IgA1P to a more distantly related metallo-IgA1P from the opportunistic species *Gemella haemolysans* (*G. haemolysans*).

There are several considerations in identifying metallo-IgA1Ps. For example, while metallo-IgA1P from *S. pneumoniae*, *S. sanguis*, and *S. oralis* are highly homologous with ~70% identity[10] (Supplementary Fig. 1), the larger M26 protease class comprises more distant members beyond metallo-IgA1P (i.e., ZmpA) that do not cleave IgA1. These more distal members include ZmpB, ZmpC, and ZmpD that are ~30% identical to their metallo-IgA1P counterparts with the substrates of most of these members remaining largely uncharacterized or with conflicting data as to their particular substrate targets[7,11]. However, the genus *Gemella* encodes for potential metallo-IgA1Ps that exhibit ~50-60% identity to the previously characterized *S. pneumoniae* IgA1P[12]. In fact, a strain of *G. haemolysans* has been shown to harbor IgA1P enzymatic activity based on its ability to cleave human IgA1[13,14]. Despite *Gemella* strains being part of the normal human microbiota, *G. haemolysans* is becoming an emerging pathogen that causes meningitis and endocarditis[12,15–17]. More recently, *G. haemolysans* has been implicated in cases of bacteremia in elderly patients, indicating that *G. haemolysans* infections are more widespread than previously thought and may therefore be a growing health problem[18].

Here, we cloned, purified, and characterized a metallo-IgA1P from *G. haemolysans* in order to determine whether this more distant M26 family member subscribes to the same gating mechanism as that of *S. pneumoniae* IgA1P. This is also the first protein cloned and characterized from *G. haemolysans*. When compared to the *S. pneumoniae* IgA1P, our structures on the *G. haemolysans* IgA1P in both substrate-free and substrate-bound states provide an understanding of the conserved nature of substrate engagement within the M26 metallo-IgA1P family and the potential steps necessary for these enzymes to engage their shared substrate. Specifically, our findings reveal that despite the substrate-free *G. haemolysans* IgA1P structure showing a more open active site relative to that of the *S. pneumoniae* IgA1P, substrate binding is still necessary to further open the active site to facilitate access to

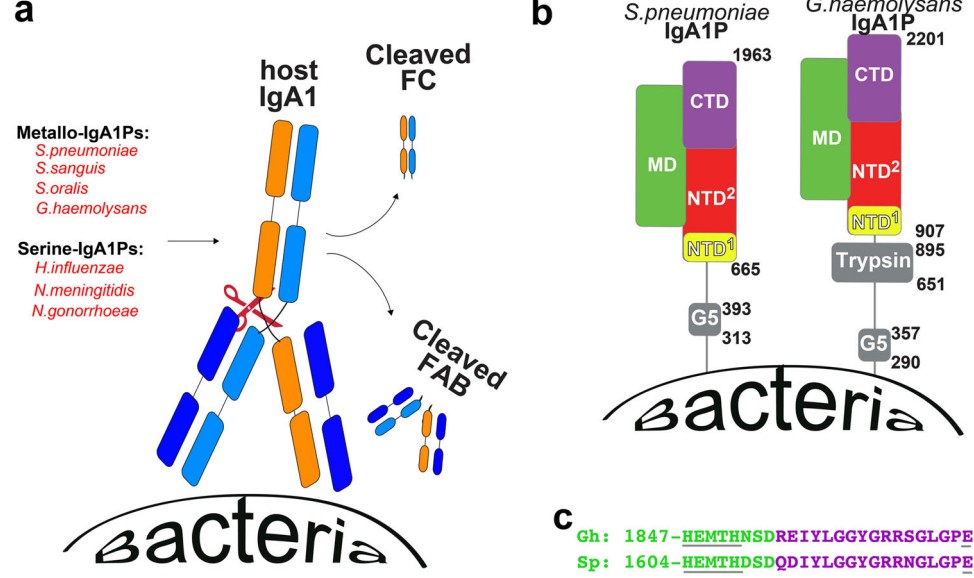

**Fig. 1 Bacterial IgA1P function and architecture of metallo-IgA1Ps. a** Cartoon representation of IgA1P function. IgA1Ps include both metallo-IgA1Ps found in Gram-positive bacteria and serine-IgA1Ps found in Gram-negative bacteria, which all target host IgA1. The IgA1 LC (navy blue) is colored similarly while the IgA1 HC is colored differently (orange, light blue) in order to delineate both bound subunits of the IgA1-FC in later figures. **b** Cartoon schematic of both the *S. pneumoniae* IgA1P and G. haemolysans IgA1P that share a similar metalloprotease region with multiple domains. These include NTD[1] (*yellow*), NTD[2] (red), MD (green), and the CTD (purple) with other domains that include the G5 and GhTrp domains also shown (gray). Residue positions for domain are indicated. For *G. haemolysans*, NTD[1]: residues 907-1024, NTD[2]: residues 1025-1340, MD: residues 1341-1854. CTD: residues 1855-2201. *For S. pneumoniae* IgA1P: NTD[1]: residues 665-781, NTD[2]: residues 782-1100, MD: residues 1101-1611, CTD: residues 1612-1963. **c** The conserved metalloprotease motif within both *S. pneumoniae* (Sp) and *G. haemolysans (Gh)* IgA1Ps includes the motif of HEMTH and downstream E that is made at the junction between the MD (green) and CTD (purple).

**Table 1 Recombinant constructs used in this study.**

| Gh IgA1P construct | Description |
|---|---|
| Gh IgA1P residues 907-2201 | M26 protease region |
| Gh IgA1P-E1848A (Residues 907-2201) | M26 protease region catalytic mutant |
| GhTrp residues 684-896 | Minimum trypsin-like domain |
| GhTrp residues 651-896 | N-terminal extension of trypsin-like domain |
| Gh IgA1P residues 651-2201 | Trypsin-like domain with M26 protease |

The name of the constructs are listed that delineate the explicit residues (left column) with the regions that they encode (right column).

the hinge region of the shared IgA1 substrate. Thus, our structural studies together with sequence comparisons and biochemical studies suggest that the ability of metallo-IgA1Ps to cleave is completely reliant on substrate binding. Finally, we have also cloned and interrogated an additional trypsin-like domain found in *G. haemolysans* IgA1P adjacent to its M26 metallo-IgA1P region. Despite the dual presence of this trypsin-like domain and M26 metallo-IgA1P domain in multiple species, it does not contribute to IgA1 cleavage and therefore suggests an unrelated activity.

## Results

**G. haemolysans metallo-IgA1P comparisons to other family members identify specific constructs to interrogate.** *S. pneumoniae* IgA1P (Sp) and *G. haemolysans* (Gh) IgA1P exhibit similarities common to the entire metallo-IgA1P family (i.e., ZmpA, M26 family) but also differences that help guide the specific selection of recombinant constructs to interrogate in this study (Fig. 1 and Supplementary Fig. 1). These similarities and differences are described separately herein that were used as a rationale for structurally interrogating the *G. haemolysans* IgA1P M26 region alone (residues 907-2201) and the potential role of adjacent regions in IgA1 cleavage, respectively.

Regarding similarities, gram-positive metallo-IgA1Ps all share three features that include a conserved N-terminal LPXTG motif for proper localization, an N-terminal G5 domain, and a C-terminal M26 protease region all connected by unstructured regions (Fig. 1). In fact, these metallo-IgA1Ps are often referred to as either "G5-containing proteins" or "M26 proteases" based on both these conserved sequences. First, all M26 metallo-IgA1Ps comprise the N-terminal conserved LPXTG motif recognized by sortase A in order to cleave the enzyme 56 residues downstream to this site (see Supplementary Figure 1 where "X" is any amino acid) and place the mature enzyme on the outside of the membrane[19]. Such cellular attachment may be followed by cellular release, which in some cases is through further enzymatic cleavage[20] or vesicle shedding as we have observed for the *S. pneumoniae* IgA1P[21]. Second, all M26 metallo-IgA1Ps comprise at least one N-terminal G5 domain that plays a distinct role in cellular adhesion[22]. We have in fact shown that the *S. pneumoniae* IgA1P G5 domain neither interacts with the C-terminal M26 protease region nor does it contribute to IgA1 cleavage[9,21,23]. This was not surprising, as independently folded domains responsible for a variety of adhesive functions are found on many surface-exposed bacterial proteins[24]. Third, all M26 metallo-IgA1Ps comprise the M26 region itself that is responsible for IgA1 cleavage, which corresponds to *G. haemolysans* IgA1P residues 907-2201. Based on our initial biochemical and structural studies, the metallo-IgA1P region was initially dissected into an N-terminal domain (NTD), middle domain (MD), and C-terminal domain (CTD)[9]. However, the NTD can be further divided into two separately folded regions defined here as NTD[1] and NTD[2], as it is the conformational change of the NTD[2] that facilitates the gating mechanism in the *S. pneumoniae* IgA1P to

allow for substrate entry[9]. A major goal within this study was to probe *G. haemolysans* IgA1P to determine if such a mechanism of the NTD[2] conformational change is conserved. Finally, a defining feature of metalloproteases that include M26 proteases is their conserved HEXXH motif ("X" is any residue) and a downstream glutamic acid first described for thermolysin[25]. The two histidine residues of the HEXXH motif together with a conserved downstream glutamic acid coordinate a Zn ion to polarize the carbonyl group of the substrate's scissile peptide. Importantly, the catalytic glutamic acid within this motif actives a water molecule, which is *S. pneumoniae* IgA1P E1605 and *G. haemolysans* IgA1P E1848 within an HEMTH motif of both enzymes (Fig. 1c). Thus, these comparisons identify two of the main constructs to study here that include the full wild-type M26 region of *G. haemolysans* IgA1P residues 907-2201 and the identical construct with the active site point mutation of *G. haemolysans* IgA1P-E1848A (see Table 1).

The distinguishing feature between the domain architecture of the *S. pneumoniae* and the *G. haemolysans* IgA1Ps is that the latter also comprises an additional trypsin-like domain found in a subset of metallo-IgA1Ps across multiple Streptococcal species that could also contribute to host IgA1 cleavage (Fig. 1b and Supplementary Fig. 1). The *G. haemolysans* trypsin-like domain comprises residues 684-896 (herein referred to as GhTrp residues 684-896) based on sequence similarity to its nearest structurally elucidated homolog (Supplementary Fig. 2), the *Staphylococcus aureus* (*S. aureus*) epidermolytic toxin A (ETA) protease[26,27]. Thus, no Streptococcus homolog has been characterized to date. The *S. aureus* ETA is a Glu-endopeptidase that targets a human membrane protein of skin cells to promote infection of deeper tissues[28,29]. ETA-mediated hydrolysis is dependent on a 30-residue N-terminal helix extension that is thought to regulate cleavage of its host target[26]. To confirm whether this trypsin-like domain within the *G. haemolysans* IgA1P is truly a folded domain, we modeled both the predicted minimally folded domain alone (GhTrip residues 684-896) and a longer construct with an extended N-terminus that could be necessary for function based on the *S. aureus* ETA (herein referred to as GhTrp residues 651-896). Considering the large number of structurally elucidated trypsin-like domains, it is not surprising to find that multiple computationally predicted models of this GhTrp domain converged well with a 1 Å RMSD for the backbone of all secondary structural elements (Supplementary Fig. 3a). These included models built by the SWISS-MODEL[30], I-TASSER[31], and Rosetta[32], which indicates that the appropriate catalytic triad is formed by H724, D779, and S850 (Supplementary Fig. 3b). Both of these GhTrp constructs, GhTrp residues 684-896 and the N-terminally extended GhTrp residues 651-896, were recombinantly produced and interrogated by NMR. Specifically, folded domains give rise to well-dispersed resonances in NMR spectra, especially for mixed α/β structures such as trypsin-like domains and this was indeed the case for both of these GhTrp constructs as observed in $^{15}$N-HSQC spectra (Supplementary Fig. 3c). This GhTrp domain was not found to interact with the remaining metalloprotease region, as indicated by a lack of resonance perturbations upon titration of the

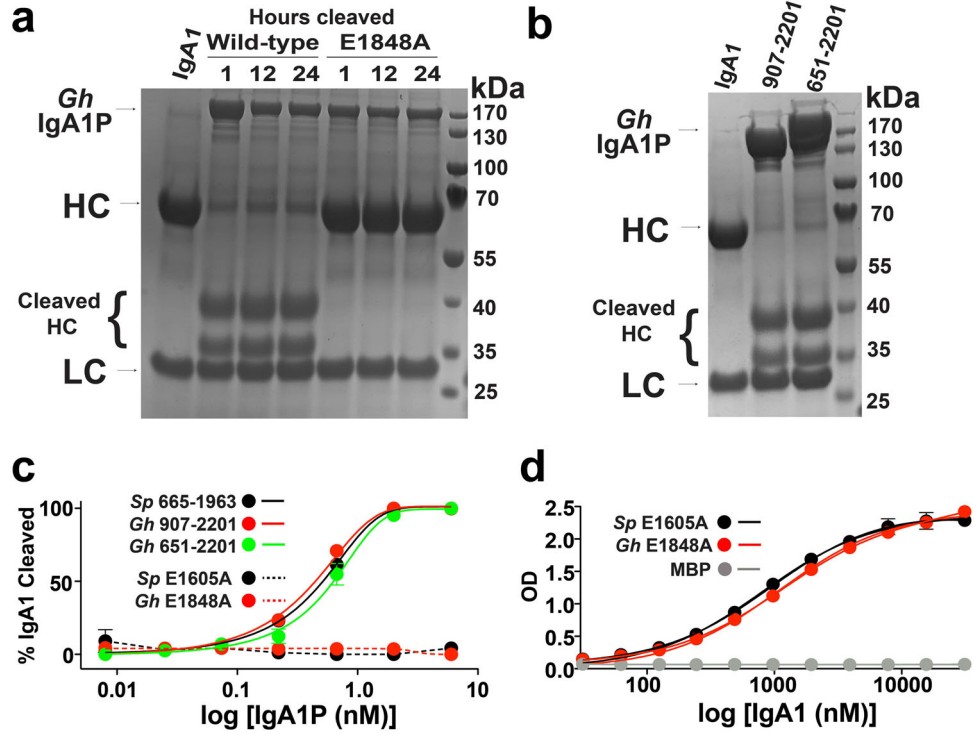

**Fig. 2 Metallo-IgA1P cleavage of IgA1. a** Proteolytic products of IgA1 after incubation with either *G. haemolysans* IgA1P WT or the E1848A mutant for the indicated time points comprised 20 µM IgA1 with 5 µM of each IgA1P. **b** Comparing the IgA1 cleavage pattern of *G. haemolysans* IgA1P with (residues 651-2201) or without (residues 907-2201) its *N*-terminal GhTrp domain. **c** Quantitative analysis of metallo-IgA1P cleavage activity of IgA1. Shown are the average of duplicate reads from an ELISA-based assay plated with the indicated construct, incubated for 30 min with IgA1, and quantified for remaining IgA1-FAB and normalized with the quantified bound IgA1-FC. These include *S. pneumoniae (Sp)* residues 665-1963 (black straight line), *G. haemolysans (Gh)* residues 907-2201 (red straight line), *G. haemolysans* residues 651-2201 (green straight line). **d** Quantitative analysis of IgA1 binding affinity to *G. haemolysans* IgA1P-E1848A and *S. pneumoniae* IgA1P-E1605A. Shown are the average of duplicate reads from an ELISA-based assay plated with 100 nM of each metallo-IgA1P mutant and incubated with the indicate concentration of IgA1.

*G. haemolysans* IgA1P residues 907-2201 (Supplementary Fig. 3d). However, whether this folded GhTrp domain contributes to IgA1 cleavage was further tested below.

**Biochemical studies reveal that metallo-IgA1Ps exhibit similar activities.** We utilized several biochemical methods to probe the activity of the *G. haemolysans* IgA1P. First, the identity of the catalytic nucleophile was confirmed by the lack of IgA1 cleavage activity of the *G. haemolysans* IgA1P E1848A point mutant (Fig. 2a). Specifically, while a time-dependent cleavage of the IgA1 substrate HC was observed within minutes for the wild-type (WT) form of *G. haemolysans* IgA1P residues 907-2201, no observable cleavage of IgA1 was observed within hours upon incubation of the *G. haemolysans* IgA1P E1848A mutant. This is consistent with the slowed cleavage of the equivalent *S. pneumoniae* IgA1P-E1605A mutant that we have previously reported[9,21]. Second, we probed the potential role of the GhTrp domain in IgA1 cleavage (Fig. 2b). There was no difference in IgA1 cleavage upon incubation of IgA1 with *G. haemolysans* IgA1P either with or without the extended GhTrp region (residues 651-2201 or 907-2201, respectively). Furthermore, neither of the purified GhTrp constructs of residues 651-894 or 684-896 cleaved the same model Boc-L-Glu-OPhenyl substrate that was cleaved by ETA[26]. Thus, analogous to the biophysical studies of the *S. aureus* ETA enzyme that proceeded identification of its substrate[28,29], the explicit function of the GhTrp remains unknown but it is not involved in IgA1 cleavage as assessed here.

Quantitative comparisons of metallo-IgA1P activity can be made by utilizing ELISA-based methods with IgA1, as the development of

kinetics assays that employ model substrates are not available, which is likely due to the specific mechanism that is reliant on recognizing IgA1 itself as described further below. The catalysis of IgA1 cleavage was compared between metallo-IgA1Ps by performing dose-response measurements (Fig. 2c). It is important to note that this assay comprises a long incubation time and does not therefore monitor the specific rate constants. Both metallo-IgA1Ps probed here exhibited similar catalytic proficiencies using this assay, with 50% IgA1 cleavage from *S. pneumoniae* IgA1P residues 665-1963 at 1.3 ± 0.3 nM and from *G. haemolysans* IgA1P residues 907-2201 at 0.8 ± 0.4 nM. The addition of the GhTrp domain did not enhance catalysis, as 50% IgA1 cleavage from *G. haemolysans* IgA1P residues 651-2201 was 2.2 ± 1.0 nM that is similar to *G. haemolysans* IgA1P residues 907-2201. Finally, in accord with the identical catalytic proficiencies of the *S. pneumoniae* IgA1P and the *G. haemolysans* IgA1P, IgA1 binding was also found to be similar with estimated dissociation constants ($K_d$) from an ELISA-based binding assay as 798±48 µM and of 1090±52 µM, respectively (Fig. 2d). We then turned our focus towards the IgA1P region itself in order to determine whether the structural basis of IgA1 engagement is conserved.

**Gating domain of substrate-free *G. haemolysans* IgA1P is partially open compared to the *S. pneumoniae* IgA1P.** We solved the structure of the *G. haemolysans* IgA1P catalytic region, *G. haemolysans* IgA1P residues 907-2201, through cryo-EM single particle reconstruction. Our purification of the intact *G. haemolysans* IgA1P residues 907-2201 resulted in a highly pure protein amenable to cryo-EM analysis (Supplementary Fig. 4). A

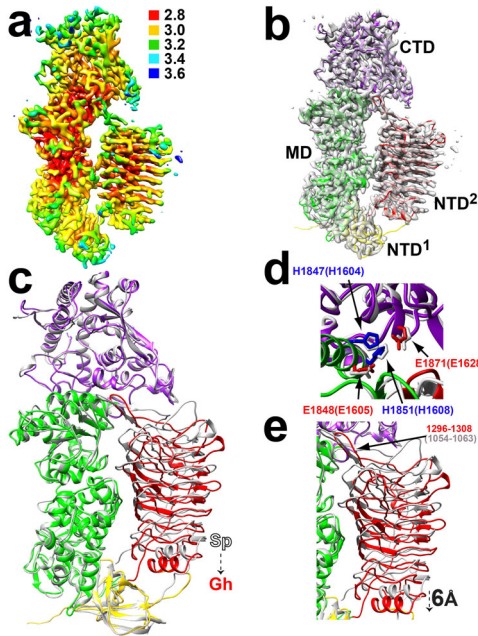

**Fig. 3 Cryo-EM 3D structure of the substrate-free *G. haemolysans* IgA1P with comparisons to of the substrate-free *S. pneumoniae* IgA1P. a** 3D single particle reconstruction of *G. haemolysans* IgA1P residues 907-2201 colored according to local resolution estimates (units in Å). **b** Density is shown from the single particle reconstruction along with the ribbon model of *G. haemolysans* IgA1P. Each domain is colored as described in Fig. 1b. **c** Structural superposition of *G. haemolysans* IgA1P residues 907-2201 (colored by domain) with *S. pneumoniae* IgA1P residues 665-1963 previously solved (white)[9]. **d** Blow-up of the active site of the conserved metalloprotease motif. The Zn-coordinating *G. haemolysans* IgA1P glutamic acid residues (red) and histidine residues (blue) are shown along with the same residues within the *S. pneumoniae* IgA1P (white). Residue numbers for the *S. pneumoniae* IgA1P are in brackets. **e** Structural comparison of the NTD[2] domains and loops that occlude the active site. The explicit 6 Å conformational repositioning of this domain between the two substrate-free metallo-IgA1Ps is highlighted. Structural alignments were conducted in Chimera using all domains of the metallo-IgA1P for a least-squares RMSD superposition.

cryo-EM 3D reconstruction of this metallo-IgA1P resulted in a 3.2 Å resolution map that readily afforded a near-atomic-resolution model that includes the active site (Fig. 3a, b). The comparison to the previously solved *S. pneumoniae* IgA1P M26 region of residues 665-1963[9], revealed both similarities and differences.

In regard to similarities that exclude the NTD[2] gating domain described further below, the *G. haemolysans* IgA1P NTD[1], MD, and CTD superimpose well with their analogous domains within the *S. pneumoniae* IgA1P with an RMSD of 1.1 Å over all Cα (Fig. 3c). The RMSD increases to 2.3 Å over all Cα atoms when including the NTD[2], which is further discussed below in regard to differences. Thus, it was not surprising to find that the active sites of these metallo-IgA1Ps are also similar. Specifically, residues comprising the conserved metalloprotease signature HEMTH motif (residues 1847-1851) and the downstream E1871 are structurally perfectly aligned with those within *S. pneumoniae* IgA1P (Fig. 3d). This suggests that the residues of the IgA1 hinge would be similarly placed within the active site between the two IgA1Ps.

The most striking difference between the two substrate-free IgA1Ps is that the *G. haemolysans* IgA1P NTD[2] gating domain is shifted to a more open position by ~6 Å relative to that of *S.*

*pneumoniae* IgA1P (Fig. 3e). The NTD[2] constitutes multiple β-strands that wind up into a twisted barrel-like structure referred to as a β-helix, which is the only shared structural fold found within these metallo-IgA1Ps to other proteins. A β-helix is widely utilized as a structural scaffold in multiple bacterial enzymes that have catalytic residues or entire domains emanating from them[33]. For example, the structure of the functionally similar *H. influenzae* serine-IgA1P also comprises such a β-helix attached to its chymotrypsin-like domain[8], although the metallo-IgA1P β-helix is approximately half as long. The large shift of the *G. haemolysans* IgA1P NTD[2] can be viewed as a partially open conformation existing along the conformational continuum between the substrate-free and substrate-bound states. For example, the gating mechanism could subscribe to an inherent opening/closing (opening/closing of the "gate"). However, this partial opening does not give any measurable advantage in binding affinity described above for the *G. haemolysans* IgA1P inactive mutant over that of its *S. pneumoniae* IgA1P counterpart. Interestingly, both metallo-IgA1Ps comprise an extended structure within their active site that are residues 1296-1308 in *G. haemolysans* IgA1P (corresponding to residues 1054-1063 in *S. pneumoniae* IgA1P), which would occlude substrate entry (Fig. 3e). Thus, while the largest difference between these first two structurally elucidated metallo-IgA1Ps is largely confined to the orientation of the NTD[2] gating domain and residues that occlude substrate binding, this does not underlie any apparent substrate binding differences.

**A gating mechanism is conserved among metallo-IgA1Ps.** The catalytically impaired mutant of *G. haemolysans* IgA1P-E1848A was used to trap the enzyme in complex with the substrate IgA1 to compare its structure to the analogous *S. pneumoniae* IgA1P-E1605A/IgA1 complex previously determined[9]. Our purification of the *G. haemolysans* IgA1P-E1848A mutant bound to mono-meric IgA1 resulted in a highly pure complex amenable to cryo-EM analysis (Supplementary Fig. 5). A cryo-EM single particle reconstruction yielded a 3.5 Å resolution map that facilitated model building (Fig. 4a, b). Analogous to the previously solved *S. pneumoniae* IgA1P-E1605A/IgA1 complex[9], the IgA1 substrate hinge region was located within the *G. haemolysans* IgA1P-E1848A active site (Fig. 4c). Only one of the IgA1-FAB fragments is visible, as only one of the two hinge regions within the intact IgA1 is cleaved at a time, as steric hinderance prevents a second IgA1P from binding the other IgA1 hinge. Although most domains within the *G. haemolysans* IgA1P bound state did not incur large changes relative to the substrate-free structure, the NTD[2] does undergo a conformational change. Specifically, the NTD[2] repositions 4 Å relative to its free form (Fig. 4d, e). Thus, while the substrate-free *G. haemolysans* IgA1P NTD[2] is already 6 Å lower than it is within the substrate-free *S. pneumoniae* IgA1P (Fig. 3e), it still must undergo a further opening to facilitate IgA1 substrate binding. These findings confirm that a gating mechanism is conserved among metallo-IgA1Ps in order to facilitate substrate binding and subsequent cleavage.

**Substrate binding induces a higher degree of structural similarity between metallo-IgA1Ps.** Despite local differences between the substrate-bound *G. haemolysans* IgA1P complex and the previously solved *S. pneumoniae* IgA1P complex, the overall structures of the enzymes are similar both within and adjacent to their active sites (Fig. 5a). Specifically, the conformational changes in NTD[2] by the gating mechanism in both metallo-IgA1P complexes results in similar conformations within their bound states that is in contrast to larger differences observed of the NTD[2] within their substrate-free states (Fig. 5b). This similarity is further illustrated by a quantitative comparison of the RMSDs

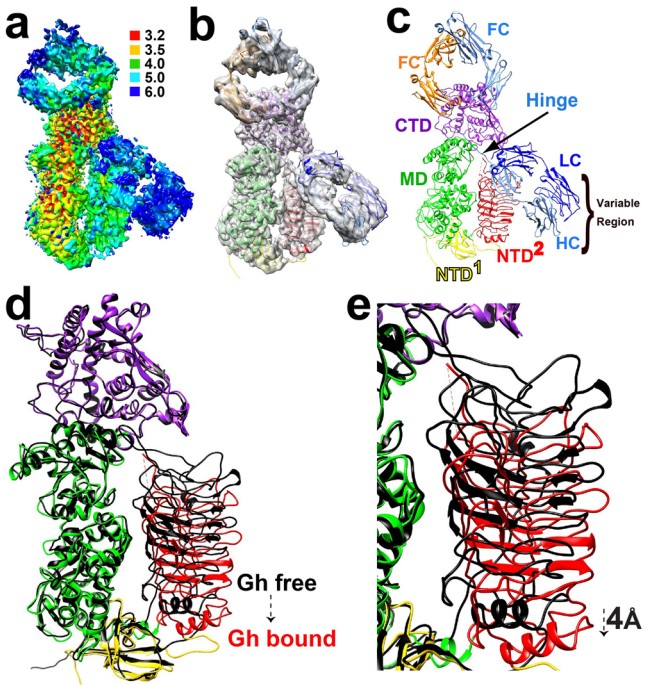

**Fig. 4 Cryo-EM 3D structure of the *G. haemolysans* IgA1P complex with IgA1 with comparisons to the substrate-free state. a** 3D reconstruction of the *G. haemolysans* IgA1P-E1848A/IgA1 complex colored according to local resolution estimates (units in Å). **b** Density is shown from the single particle reconstruction along with the ribbon model of *G. haemolysans* IgA1P-E1848A and the bound IgA1. Each domain of the enzyme and IgA1 is colored as described in Fig. 1a, b. The IgA1 HC monomers are colored light blue and orange while the IgA1 LC fragments are colored navy blue. Only one IgA1-FAB fragment is visible in the density. **c** The full *G. haemolysans* IgA1P-E1848A/IgA1 complex model with each domain specifically delineated. **d** Structural comparison of the substrate-free *G. haemolysans* IgA1P (black) and the *G. haemolysans* IgA1P-E1848A/IgA1 complex (colored by domain). **e** The 4 Å conformational repositioning of the NTD$^2$ domain and active site loops are shown between the substrate-free *G. haemolysans* IgA1P (black) and within its complex with IgA1 (colored) indicating the conformational change. Structural alignments of the substrate-free and substrate-bound complexes were conducted in Chimera using all domains of the metallo-IgA1P for a least-squares RMSD superposition.

between the two metallo-IgA1Ps whereby the majority of RMSDs within the *G. haemolysans* IgA1P NTD$^2$ collapsed to much smaller values upon IgA1 binding (Fig. 5c). Even more interestingly, the variability of the bound IgA1 substrates relative to each other is dramatically diminished both within the active site where the IgA1 HC hinge region lies and adjacent residues within the second Ig domain of the HC that includes residues 124-220 (Fig. 5d). Arguably, the most important similarity between the two bound metallo-IgA1P complexes is that the bound IgA1 hinge region and the conserved HEMTH along with the downstream glutamic acid motif are in similar positions (Fig. 5e). This is also consistent with previous studies that indicate metallo-IgA1Ps are all thought to cleave the host IgA1 hinge region of the HC between residues P227/T228[5,34]. A final similarity that is worth noting is missing density for both bound substrates within the IgA1 hinge region of the HC that is downstream to their shared cleavage sites (residues 232-241). This suggests this region is largely disordered in both bound metallo-IgA1Ps (Fig. 5f).

The primary differences between the substrate-bound metallo-IgA1Ps complexes are active site loops, which undergo a conformational change in concert with the conformational shift of the NTD$^2$

relative to their substrate-free states (Fig. 5b, f). This loop comprises *G. haemolysans* IgA1P residues 1292-1318 that is structurally synonymous with *S. pneumoniae* IgA1P 1048-1069. In fact, these loops are in different orientations within both their substrate-free and substrate-bound states, as illustrated by their RMSDs for both states (Fig. 5c). Interestingly, both metallo-IgA1Ps have similar stretches of missing density within these loops that comprise *G. haemolysans* IgA1P residues 1298-1307 and *S. pneumoniae* IgA1P residues 1054-1063 (Fig. 5f). However, in the substrate-bound *G. haemolysans* IgA1P complex this loop is much closer to the IgA1 hinge region, whereby the missing density could potentially interfere with substrate binding. As both metallo-IgA1Ps exhibit similar substrate IgA1 binding affinities and catalytic proficiencies quantified above, this could suggest that the missing densities in both metallo-IgA1P active sites may interfere in function part of the time. Therefore, we significantly shortened this loop region by 10 residues within *G. haemolysans* IgA1P that largely removed its disordered region to produce the deletion mutant of Δ1297-1306. *G. haemolysans* IgA1P Δ1297-1306 exhibited the same catalytic proficiency as the WT enzyme (Fig. 5g), indicating that these disordered residues do not significantly impact IgA1 cleavage within this assay. Taken together, the specific differences between the first two metallo-IgA1Ps solved appear to be less important than their similarities that are especially evident within their bound forms.

## Discussion

Despite the discovery of IgA1Ps nearly a half a century ago[35], the mechanistic details of their interaction with their IgA1 substrate that underlies bacterial immune evasion has largely remained unknown until recently. Our combination of biochemical studies and cryo-EM single particle reconstructions of the *G. haemolysans* IgA1P here, together with our recent studies on the *S. pneumoniae* IgA1P[9,21], has revealed a conserved gating mechanism for the metallo-IgA1Ps with further implications for substrate binding described herein. First, the metallo-IgA1P gating mechanism is reliant on the NTD$^2$ domains, which form β-helix structures that partially occlude the active sites and must open for proper placement of the IgA1 substrate hinge region (Fig. 6a). Despite the surprising difference between the positioning of their NTD$^2$ domains (Fig. 6a, left two structures), both metallo-IgA1Ps must open to the same extent in order to facilitate substrate binding (Fig. 6, right two structures). Second, the conserved metallo-IgA1P mechanism appears to be reliant on the substrate itself for active site gating. This suggests that metallo-IgA1P binding to their IgA1 substrate at least partially subscribe to an "induced fit" model, as the active site is occluded by the NTD$^2$ domain in their substrate-free structures[36,37]. Interestingly, our biochemical binding data reveal that the partial opening of the *G. haemolysans* IgA1P relative to the *S. pneumoniae* IgA1P does not lead to any advantage in IgA1 binding affinity. Such a dependence of metallo-IgA1Ps on their exact substrate also explains the complete lack of peptide mimics identified for this family. In stark contrast to metallo-IgA1Ps, model peptide substrates have been identified for the *H. influenzae* serine-IgA1P, which may subscribe to a "conformational selection model" proposed for other trypsin-like domains[38–40]. However, it is difficult at this stage to completely assess the particular catalytic models of metallo-IgA1Ps without a catalytic assay that is sensitive to the microscopic kinetic rates of cleavage. For example, our catalytic assessments only provide the overall proficiencies under these particular conditions that include an incubation step and would therefore encompass many turnover events that do not provide insight into the rate-limiting steps of turnover. Finally, data provided here also implicate the relative importance of substrate interactions. For example, conserved regions may be expected to underlie the conserved binding

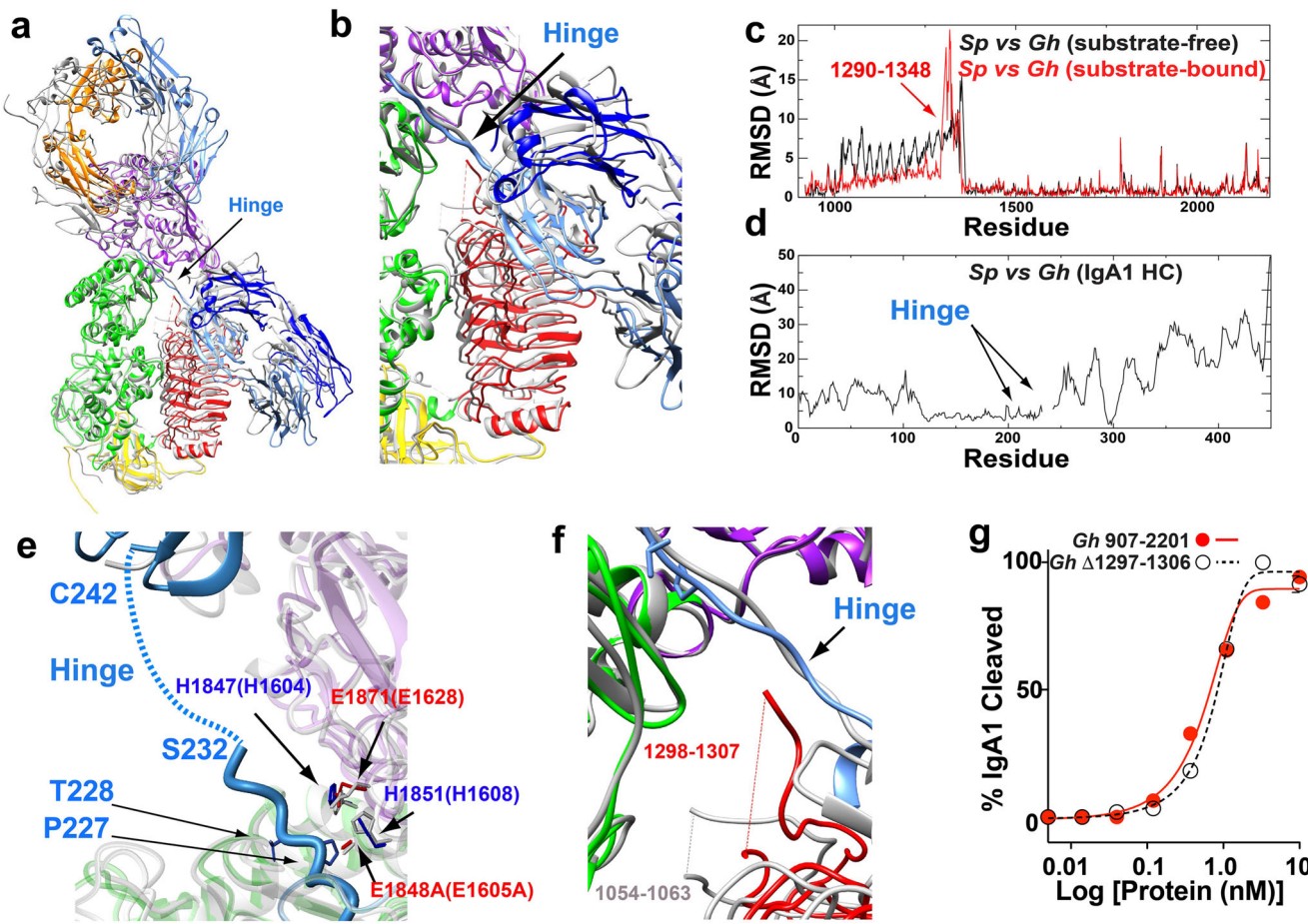

**Fig. 5 Structural comparisons between *G. h*aemolysans IgA1P and S. pneumoniae IgA1P complexes with IgA1. a** The *G. haemolysans* IgA1P-E1848A/ IgA1 complex is shown (colored) superimposed on the *S. pneumoniae* IgA1P-E1605A/IgA1 complex (white). **b** Blow-up of the region that includes the NTD[2] domains and active site loops. **c** RMSDs between the two metallo-IgA1Ps in their substrate free (black) and substrate-bound state (red). **d** RMSDs between the two bound IgA1 HCs within their respective metallo-IgA1P. **e** Blow-up of the active sites with the conserved metalloprotease motif in the bound states. The *G. haemolysans* IgA1P glutamic acid residues (red) and histidine residues (blue) are shown along with their counterparts within the *S. pneumoniae* IgA1P (white). **f** Blow-up of the active site loops show missing density in the IgA1-bound *G. haemolysans* IgA1P-E1848A for residues 1298-1307 and *S. pneumoniae* IgA1P-E1605A for residues 1054-1063. **g** Catalytic proficiencies are compared between *G. haemolysans* IgA1P and the *G. haemolysans* IgA1P **Δ**1297-1306 mutant.

affinities of the two metallo-IgA1Ps. To this point, the surface residues of the *G. haemolysans* IgA1P that interact with the IgA1-FAB are relatively well conserved within the *S. pneumoniae* IgA1P compared to the reaction surface with the IgA1-FC (Fig. 6b). These include much of the exposed loop within residues 1064-1069 within the metallo-IgA1P CTD and multiple β-strands within the NTD[2]. Thus, metallo-IgA1P binding may first comprise IgA1 engagement through these conserved regions of the IgA1-FAB followed by less specific interactions with the IgA1-FC that induce active site opening (i.e., gating). A higher specificity of the IgA1Ps with IgA1-FAB may also be consistent with the lower RMSDs between the bound metallo-IgA1P and their bound IgA1. However, as the cleaved IgA1 products do not co-migrate with the enzyme, as previously reported for *S. pneumoniae* IgA1P[21], both IgA1-FAB and IgA1-FC interactions are still necessary for a high affinity interaction.

Structural elucidation and biochemical interrogations of the first two metallo-IgA1Ps studied here also have implications for other proteins beyond these virulence factors. For example, metallo-IgA1Ps (otherwise known as ZmpA proteases) are among the M26 class of proteases that also include ZmpB, ZmpC, and ZmpD[7], yet their specific substrate interactions remain poorly characterized. While both metallo-IgA1Ps (ZmpA) and ZmpB are present in

multiple Streptococcus species and all strains of *S. pneumoniae*, the presence of both ZmpC and ZmpD are highly variable[41]. ZmpB has been shown to be a virulence factor for *S. pneumoniae* infection in mice models and is associated with collagen[42-44]. ZmpC correlates with clinical severity of *S. pneumoniae* infection and activates matrix metalloprotease-9 (MMP-9)[7,45]. However, a direct interaction for any of these other M26 protease family members and their proposed substrates has yet to be shown. Considering the relatively high affinity of both the *S. pneumoniae* and *G. haemolysans* IgA1P metallo-IgA1P active site mutants for their IgA1 substrate that we have now determined, such studies suggest that similar inactive mutants can be produced for these other Zmp family members to broadly pull-down their interacting partners (without cleaving them). Such studies can then be used to determine whether similar gating mechanisms are also utilized by these M26 proteases for their cognate substrates. Another interesting finding that may have further implications is the fact that we confirmed the GhTrp domain is well-folded. Although this domain did not have any cleavage activity towards IgA1, whether such domains serve other purposes remains unknown. Interestingly, there is precedent to suggest that bacterial surface expression of enzyme-like modules does not always result in a catalytic role. For example, surface exposed proteins may instead serve very different roles that include adhesion described for

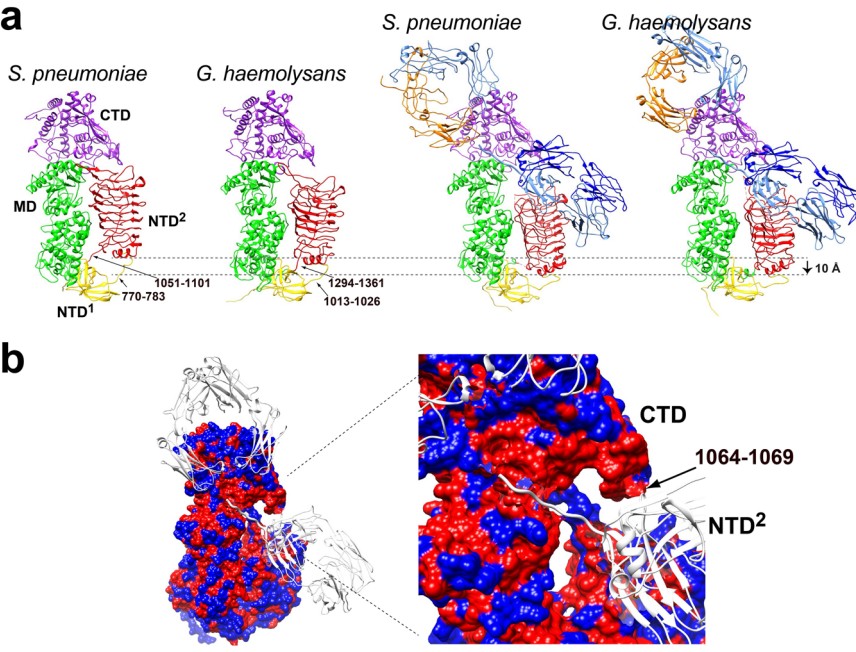

**Fig. 6 Comparisons of the metallo-IgA1P gating mechanisms. a** Both the substrate-free and substrate-bound *G. haemolysans* IgA1P and *S. pneumoniae* IgA1P models are shown with the specific linkers that connect the NTD² domains. The dashed line defines the conformational gate required for opening ~10 Å for *S. pneumoniae* IgA1P and ~4 Å for *G. haemolysans* IgA1P in order to allow IgA1 substrate binding. **b** The structure of *G. haemolysans* IgA1P-E1848A/IgA1 is shown with the IgA1P colored according to conservation (red for conserved, blue for unconserved). The bound IgA1 is also shown (white). The blow-up (right) shows the IgA1 substrate binding pocket that includes residues within the NTD² and CTD domains with conserved residues with residues 1064-1069 also delineated.

the *S. pneumoniae* β-galactosidase[46]. Finally, it should be noted that many of the antibiotic resistant strains recently listed by the World Health Organization as "high-priority" encode their own IgA1P[47]. The replacement of vaccines that target strain-specific sugar antigens with more widely conserved proteins such as these M26 family members here has recently been proposed[48]. Thus, our structural elucidations of the first two ZmpA family members that includes the previous *S. pneumoniae* IgA1P catalytic region[9] and the *G. haemolysans* IgA1P complexes studied here may be used to identify both conserved and exposed regions such as the CTD and several regions within the NTD² responsible for IgA1 engagement.

## Methods

**Plasmid constructions**. All inserts of *G. haemolysans* IgA1P were PCR amplified from chromosomal DNA (Gha_ATCC10379) obtained from the Leibniz Institute DSMZ – German Collection of Microorganisms and Cell Cultures GmbH (Braunschweig, Germany). PCR products coding for *G. haemolysans* IgA1P residues 907-2201 and 651-2201 were inserted into a pET21b vector encoding for an N-terminal MBP, a thrombin cleavage site, a BamH1 site for PCR product insertion, and a C-terminal 6xHis tag using the single BamH1 site. The *G. haemolysans* IgA1P E1848A mutation and the Δ1297-1306 deletion mutant within the construct containing 907-2201 was produced by first PCR amplifying a 3' PCR fragment using the primers that included the required mutation and then utilizing this PCR fragment as the 3' primer specific for the N-terminal fragment of 907-2201 for a second reaction to produce the full 907-2201 product with the encoded mutation. PCR products for the GhTrp domain that included 651-896 and 684-896 were inserted into pET21b with an N-terminal 6xHis tag and thrombin cleavage site using the single NdeI site. CloneAmp HiFi PCR mix (Takara) was used for all PCR reactions and ligase-independent cloning using In-Fusion (Takara) was used for ligation-independent cloning. All plasmids were confirmed by sequencing. The production of *S. pneumoniae* IgA1P and its mutant have been previously described[9].

**Protein expression and purification**. *E. coli* BL21(DE3) cells with transformed plasmids were grown at 37 °C and induced at an OD₆₀₀ of 0.6–0.9 with 1 mM IPTG for 3 h before harvesting. Typical growths comprised 4 L of Luria Broth (LB) supplemented with ampicillin for the pET21 vectors utilized. For ¹⁵N-labeled proteins, M9 minimal media was used (6 g/l Na₂HPO₄, 3 g/l KH₂PO₄, 0.5 g/l NaCl, 1 g/l NH₄Cl, 2 g/l glucose, 2 ml of 1 M MgSO₄ 100 ml of 1 M NaCl CaCl₂, 10 mg/l

thiamine) with all other grown conditions identical to unlabeled growths in LB. Cells were lysed via sonication and centrifuged for 15 minutes prior to applying supernatants to their initial Ni columns (Sigma). All purifications were conducted on an AKTA prime FPLC system (GE Healthcare).

Two sets of purifications were performed that included one for the pET21-MBP-IgA1P-6xHis plasmids and one for the pET21-6xHis-GhTrp plasmids. For the pET21-MBP-IgA1P-6xHis purifications, supernatants were applied to a Ni affinity column using Ni buffer (50 mM phosphate buffer, pH 7.0, 500 mM NaCl, and 10 mM imidazole) and eluted using Ni buffer supplemented with 400 mM imidazole. Ni affinity elutions were then applied to an amylose column (Cytiva) using amylose buffer (20 mM Tris, pH 7.4, and 200 mM NaCl) and eluted using 20% (w/v) glucose in amylose buffer. The eluent was then concentrated, cleaved with thrombin (Sigma) at room temperature overnight, and diluted for reapplication to amylose resin to strip the remaining MBP. Finally, amylose-resin flow-through was concentrated and applied to an S200 size exclusion column (120 ml, Cytiva) equilibrated in 20 mM HEPES, pH 7.4, and 300 mM NaCl. For the pET21-6xHis-GhTrp purifications, lysed supernatants were also applied first to a Ni affinity column using the identical buffers described above. Eluants were concentrated, cleaved with thrombin overnight at room temperature, and applied to an S75 size exclusion column (120 ml, Cytiva) equilibrated with 50 mM HEPES, pH 7.0, and 150 mM NaCl.

Human IgA1 with kappa LC was purchased commercially (myBioscience; San Diego, CA), but was further purified over a preparative S200 size exclusion column (120 ml, Cytiva) in 20 mM HEPES, pH 7.4, and 150 mM NaCl in order to specifically select for monomeric IgA1.

**Biochemical analyses**. To compare catalytic proficiencies via ELISA-based assays, the indicated proteins were first incubated with 10 µg/ml IgA1 at 37 °C for 1 h in 25 µl total volumes. Intact versus cleaved IgA1 was quantified by first coating a Nunc MaxiSorp plate (ThermoFisher; Waltham, MA) with 1 µg/ml goat anti-human IgA CH3 (Novus; Wrentham, MA). Plates were blocked with PBST-0.5% BSA for two hours and the 25 µl of each reaction was diluted 10-fold and added to each well for an additional 2 h at room temperature and then washed three times with PBST. Intact or cleaved IgA1 was detected by the presence or absence of bound kappa light chain using alkaline phosphatase-labeled goat anti-human Kappa (Sigma-Aldrich). IgA1 binding was normalized by detecting bound IgA1 heavy chain with HRP-labeled goat anti-human Fcα (Jackson ImmunoResearch). Plates were either developed with p-nitrophenyl phosphate (PnPP) substrate-AP (Sigma) or ABTS-HRP and read on a Versamax plate reader (Molecular Devices).

To quantify binding of IgA1 to the metallo-IgA1P mutants of *G. haemolysans* IgA1P-E1948A and *S. pneumoniae* IgA1P-E1605A, 20 nM of each protein was first

**Table 2 Cryo-EM data collection, refinement and validation statistics for G. *haemolysans* IgA1P res. 907-2201 and the IgA1P-E1848A mutation in complex with IgA1.**

|  | IgA1P | IgA1P + IgA1 |
|---|---|---|
| Data Collection and Processing |  |  |
| Microscope | Titan Krios | Titan Krios |
| Voltage (kV) | 300 | 300 |
| Magnification (nominal) | 105,000 | 29,000 |
| Electron Dose (e⁻/Å²) | 46 | 49 |
| Camera | Gatan K3 BioQuantum | Gatan K3 |
| Defocus range (um) | −0.7 to −2.0 | −0.8 to −2.3 |
| Pixel size (Å) super resolution | 0.415 | 0.401 |
| Movies collected | 6,367 | 11,568 |
| Symmetry imposed | C1 | C1 |
| Final particle images (no.) | 443,908 | 205,808 |
| Map resolution (Å) | 3.25 | 3.53 |
| Sharpening B-factor (Å²) | 160.2 | 109.4 |
| Software used to process data | cryoSPARC | cryoSPARC |
| Refinement statistics |  |  |
| Number of protein atoms (non-H) | 10,350 | 16,508 |
| R.m.s. deviations | – | – |
| Bonds (Å) | 0.007 | 0.008 |
| Bond angles (°) | 1.020 | 1.006 |
| Validation | – | – |
| MolProbity score | 2.09 | 2.92 |
| Clash score | 9.63 | 13.35 |
| Poor rotamers (%) | 1.02 | 0 |
| Ramachandran plot | – | – |
| Favored (%) | 91.59 | 94.82 |
| Allowed (%) | 8.41 | 5.18 |
| Disallowed (%) | 0 | 0 |
| C-beta deviations | 0 | 0 |
| Model vs Data CC | 0.83 | 0.82 |
| FSC model (0.143) | 3.7 | 3.8 |
| EMDB access code | EMD-26812 | EMD-26813 |
| PDB access code | 7UVK | 7UVL |

motion corrected using MotionCor2[50] and their contrast transfer functions were estimated using Gctf[51]. Meaningful representative class averages were selected and used for template picking. The particles were extracted with a box size of 336 ×336 pixels and subjected to 2D classification. Classes with clear visible secondary features were selected and subjected to ab initio reconstruction, 3D classification, and refinement. The resolution of the final maps was based on the gold-standard Fourier shell correlation (FSC) measurement. Initial models for both substrate-free and the substrate-bound complex were built in Chimera[52], which used the previously determined structure of the *S. pneumoniae* IgA1P and IgA1 as a template[9]. Models were then refined in Coot[53] and Phenix[54].

**Reporting summary**. Further information on research design is available in the Nature Research Reporting Summary linked to this article.

## Data availability

The cryo-EM maps are deposited in the Electron Microscopy Data Bank under accession codes "EMD-26812" (*G. haemolysans* IgA1P 907-2201), "EMD-26813" (*G. haemolysans* IgA1P-E1848A 907-2201 in complex with IgA1). Structure coordinates are deposited at the Protein Data Bank with accession codes "7UVK" and "7UVL", respectively.

coated onto a Nunc MaxiSorp plate. Plates were washed and blocked for 2 h at room temperature. Plates were washed again prior to adding human IgA1 with kappa LC for 2 h, washed three times with PBS, and anti-human IgA HRP was added for one hour at room temperature. Finally, plates were washed and developed using ABTS Peroxide and read on a Veramax plate reader to obtain ODs.

**NMR sample preparation and data collection**. All NMR data were collected on a Varian 900 equipped with a cryoprobe. Both ¹⁵N GhTrp samples of 651-896 and 684-896 were produced as described above with all NMR samples concentrated directly after the final S75 column to 500 μM with 20 μl of D₂O in order to lock.

**Cryo-EM sample preparation and data collection**. For the *G. haemolysans* IgA1P residues 907-2201 alone, purified protein was concentrated to 340 μM (~5 mg/ml) in 20 mM HEPES, pH 7.4, and 300 mM NaCl. For the *G. haemolysans* IgA1P-E1848A/IgA1 complex, purified IgA1P-E1948A (residues 907-2201) was concentrated to 100 μM and incubated with 30 μM of the IgA1 substrate in the presence of 1 mM EDTA. This *G. haemolysans* IgA1P/substrate complex was further purified in 20 mM HEPES, pH 7.4, and 300 mM NaCl, 1 mM EDTA over an analytical Superdex 200 (Cytiva) in order to enrich for the 1:1 complex (Supplementary Fig. 1d). Fluoro-octyl maltoside (FOM, Anatrace) was added 20 μl at a final concentration of 0.7 mM of either IgA1P alone or in complex prior to freezing. For grid preparations, 3 μl of ~6 mg/ml sample was applied to plasma-cleaned C-flat holy carbon grids (1.2/1.3, 400 mesh) and frozen using a Vitrobot Mark IV (Thermo Fisher Scientific), with the environmental chamber set at 100% humidity and 4 °C. The grids were blotted for 2.5~3.0 s and then flash frozen in liquid-nitrogen-cooled liquid ethane. A full description of the cryo-EM data collection can be found in Table 2.

**Cryo-EM data processing and structural modeling**. All data for cryo-EM 3D single particle reconstructions were processed in cryoSPARC[49]. Movies were

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

## Acknowledgements

We thank the CU Cryo-EM Structural Biology Shared Resource Facility for screening. Data collection for single particular reconstructions were collected at the Pacific Northwester Cryo-EM Center (PNCC) at Oregon Health and Science University (OHSU), supported by NIH grant U24GM129547 and accessed through EMSL (grid.436923.9), a DOE Office of Science User Facility sponsored by the Office of Biological and Environmental Research. H.Z. was supported by NIH R01 GM126626. E.Z.E. was supported by NIH R21 AI146295 and R01 GM139892.

## Author contributions

E.Z.E. and H.Z. determined the 3D reconstructions and E.Z.E., J.S.R., and E.L. refined structures. N.M. collected cryo-EM data. J.R., T.H., and N.T. performed biochemistry experiments. A.J.M.-G. provided guidance on molecular engineering of constructs. J.S.R. and E.Z.E. purified all proteins and complexes.

## Competing interests

The authors declare no competing interests.
