## [Peer Review File · Communications Biology]

Reviewers' comments:

Reviewer #1 (Remarks to the Author):

Following up with the previous analysis of IgA1 metalloprotease from *S. pneumoniae* (IgA1P), Redzic et al use biochemical and cryoEM analysis to study a homologous zinc metalloprotease from *G. haemolysans* (Gh_IgA1P). In comparison with IgA1P, Gh_IgA1P has the additional trypsin-like domain while the protease core and surrounding N- and C-terminal domains are highly similar to each other (~50-60% identity). They show that the trypsin domain of Gh_IgA1P is stable based on the NMR study but does not contribute to IgA1P ability to degrade IgA. They then solved the cryoEM structure of Gh_IgA1P protease core and surrounding N- and C-domain. They found the IgA1 bound structure of this enzyme is near identical to that of IgA1P while the apo-enzyme has a small but significant rigid body motion in the N-terminal domain next to the catalytic core. They conclude that this family of IgA1 proteases uses a conserved "induced fit" mechanism to adapt the binding of IgA1 in order to selectively bind and inactivate IgA1. Overall, the analysis is well executed and the conclusion is sound even though the novelty is not particularly high. In addition, several concerns should be addressed.

1. CryoEM data is not well presented and the analysis can be better executed. For example, there is no report to address the orientation bias, particularly in the light of the poorer map quality than the expected reported resolution. Furthermore, the exemplary 2D class images are of poor quality and only three are shown. In addition, there is still 0.47% residues in Ramachandran disallowed region. Given the size of the protein, it represents substantial residues in the disallowed region, suggesting that a substantial structure refinement remains to be done.
2. The authors strongly suggest that their work provides the basis for the development of broad-based vaccine against multiple pathogens, throughout the manuscript without any good explanation. It is really unclear what is the key additional insight that this work has provided in comparison with the previous structural work and the sequence analysis. The authors should either provide additional support to this notion or tune down such argument.

Minor issues: Supplemental figure 4e: GhIgA1p peak 1 should be labeled IgA1 bounded Gh IgA1P then use the arrow to point to SDS page gel. Currently, the arrow could be mistakenly seen to indicate that the peak is the Gh IgA1P.

Reviewer #2 (Remarks to the Author):

The paper "A substrate-induced gating mechanism is conserved among Gram-positive IgA1 metalloproteases" by Redzic et al provides novel cryo-EM structures of a metallo IgA1 protease as found in Gram-positive bacteria, in this case of *Gemella haemolysans*. The strategy follows a template that was used by the same group to solve the IgA1P structure from *S. pneumoniae* published in Nature communications in 2020.

Overall the paper yields interesting data; it finetunes our knowledge on how IgA1 metalloproteases bind and cleave human IgA and the gating mechanism to bind to their substrates. It also provides a glimpse of an extra trypsin domain present in the *Gemella* IgA1P that was not present in the streptococcal one, although its role remains elusive. Overall the data also look convincing, but I do have some questions on the way it is presented and how some experiments are interpreted.

1. The authors rather describe their results as if they have purified and structurally elucidated a complete IgA1P (see line 85), but in fact work with domains of the complete protein. This should be better explained. For example, the cryo-EM studies appear based upon region 907-2201, which lacks some of the domains discussed in para 1 of the Results (G5 and, more significantly, the trypsin-like domain). In principle, the absence of these domains could have an influence on the overall structure

and, more likely, spatial organization of the proteins. This is important to mention and discuss, also in view of the observed differences when the substrate is absent.

2. The first paragraph is headlined as describing sequence alignments, but it also includes structural data and discussions thereof. This is confusing. Firstly, the discussion on the structure of the G5 domain may be better placed in the discussion (also in view of the fact that it was not solved here). Secondly, the discussion on the trypsin domain, based upon structure predictions and NMR studies appear out of place and better placed in a separate paragraph. This section also lacks any introductory information; NMR was performed on two constructs and I do not see an explanation why these two are chosen, nor do I see whether the NMR actually provides support for the theoretical models. I think this data could be interesting, but then needs a better introduction in a separate paragraph. Finally, the proteins appear not active on a possible substrate (see next para), so is the protein purified, indeed, the correct conformation? And then based upon what assumptions? In the discussion (l. 337) it is mentioned that the trypsin domains was well-folded, but I miss a discussion on what assumption this claim is based.

3. The biochemical analysis of paragraph two is convincing in showing the catalytic activity of the purified domains. It lacks, however, a description why the E1848 was chosen for mutation into an inactive mutant. And I have to presume the same fragment of IgA1P was tested?

minor points:

4. In the abstract the authors claim stark differences, this seems to be overstated. The Gemella IgA1P is more open when no substrate is around, but shows a quite similar architecture to the streptococcus one when substrate is bound. This is corroborated by the very similar biochemical data that show pretty similar binding properties.

5. How widely the trypsin domain in Gemella IgA1P distributed among other species?

6. the authors mention in the introduction and discussion ZmpB-D as family members if the metallo IgA1P proteases, but it remains unclear in what type of bacteria these are found, whether they exist besides the IgA1P and what the current study actually means for understanding these proteins.

7. line 160; IgA1 substrate HC; should this not be IgA1P substrate HC?

We wish to thank both of the reviewers for their kind remarks and we provide below a point-by-point response (in yellow) to their requests. We have also marked up (in yellow) the specific changes to the manuscript and within the supplementary material.

Reviewer #1 (Remarks to the Author):

Following up with the previous analysis of IgA1 metalloprotease from *S. pneumoniae* (IgA1P), Redzic et al use biochemical and cryoEM analysis to study a homologous zinc metalloprotease from *G. haemolysans* (Gh_IgA1P). In comparison with IgA1P, Gh_IgA1P has the additional trypsin-like domain while the protease core and surrounding N- and C-terminal domains are highly similar to each other (~50-60% identity). They show that the trypsin domain of Gh_IgA1P is stable based on the NMR study but does not contribute to IgA1P ability to degrade IgA. They then solved the cryoEM structure of Gh_IgA1P protease core and surrounding N- and C-domain. They found the IgA1 bound structure of this enzyme is near identical to that of IgA1P while the apo-enzyme has a small but significant rigid body motion in the N-terminal domain next to the catalytic core. They conclude that this family of IgA1 proteases uses a conserved "induced fit" mechanism to adapt the binding of IgA1 in order to selectively bind and inactivate IgA1. Overall, the analysis is well executed and the conclusion is sound even though the novelty is not particularly high. In addition, several concerns should be addressed.

Reviewer#1: CryoEM data is not well presented and the analysis can be better executed. For example, there is no report to address the orientation bias, particularly in the light of the poorer map quality than the expected reported resolution. Furthermore, the exemplary 2D class images are of poor quality and only three are shown. In addition, there is still 0.47% residues in Ramachandran disallowed region. Giving the size of the protein, it represents substantial residues in the disallowed region, suggesting that a substantial structure refinement remains to be done.

Answer: We apologize for not providing more details that we have now included for both the free Gh-IgA1P (Supplemental Figure 4) and for the complex (Supplemental Figure 5), which includes the specific steps taken in data processing, multiple 2D class averages, and the direction distribution. In addition, we have further refined the Gh-IgA1P/IgA1 complex that now comprises ZERO residues in the Ramachandran disallowed region and have replaced this in the RCSB.

2. The authors strongly suggest that their work provides the basis for the development of broad-based vaccine against multiple pathogens, throughout the manuscript without any good explanation. It is really unclear what is the key additional insight that this work has provided in comparison with the previous structural work and the sequence analysis. The authors should either provide additional support to this notion or tune down such argument.

Answer: We apologize for over-emphasizing the potential use of these structural studies for "broad based" vaccine development, which we had mentioned three times (Abstract, Introduction, and Discussion). Considering the sequence conservation within the CTD (that is exposed), we had simply meant to describe that such a region could be used for "broader" vaccines that may inhibit host IgA1 engagement. However, we have now described the potential use of these proteins as vaccines only one time in the Discussion and will allow others to make specific comparisons during any future vaccine considerations.

Minor issues: Supplemental figure 4e: GhIgA1p peak 1 should be labeled IgA1 bounded Gh IgA1P then use the arrow to point to SDS page gel. Currently, the arrow could be mistakenly seen to indicate that the peak is the Gh IgA1P.

Answer: We apologize for this lack of clarity, as the arrow was meant to point towards the SDS-PAGE gel. We have now explicitly included the "Gh IgA1P/IgA1 complex" delineating this eluted peak.

Reviewer #2 (Remarks to the Author):

The paper "A substrate-induced gating mechanism is conserved among Gram-positive IgA1 metalloproteases" by Redzic et al provides novel cryo-EM structures of a metallo IgA1 protease as found in Gram-positive bacteria, in this case of *Gemella haemolysans*. The strategy follows a template that was used by the same group to solve the IgA1P structure from *S. pneumoniae* published in Nature communications in 2020.

Overall the paper yields interesting data; it finetunes our knowledge on how IgA1 metalloproteases bind and cleave human IgA and the gating mechanism to bind to their substrates. It also provides a glimpse of an extra trypsin domain present in the *Gemella* IgA1P that was not present in the streptococcal one, although its role remains elusive. Overall the data also look convincing, but I do have some questions on the way it is presented and how some experiments are interpreted.

Reviewer #2: The authors rather describe their results as if they have purified and structurally elucidated a complete IgA1P (see line 85), but in fact work with domains of the complete protein. This should be better explained. For example, the cryo-EM studies appear based upon region 907-2201, which lacks some of the domains discussed in para 1 of the Results (G5 and, more significantly, the trypsin-like domain). In principle, the absence of these domains could have an influence on the overall structure and, more likely, spatial organization of the proteins. This is important to mention and discuss, also in view of the observed differences when the substrate is absent.

Answer: We apologize for any confusion, as we completely agree and that is specifically why we have compared the M26 region alone (Gh IgA1P residues 907-2201) along with the adjacent trypsin-like domain (Gh IgA1P residues 651-2201). This comparison of IgA1 cleavage is shown in Figure 2b,c, detailing that the trypsin-like domain (i.e., GhTrp) does not contribute to this substrate cleavage. Moreover, we performed a specific titration with NMR to assess whether this trypsin-like domain interacts with the M26 region in Supplementary Figure 3d, which revealed no interaction. Considering the lack of clarity of these details, we have re-written the first section that was renamed with the explicit point of this section indicated by the reviewer as, "*G. haemolysans* metallo-IgA1P comparisons to other family members reveal specific constructs to interrogate". Moreover, we have included a table of all of the constructs interrogated within this study to further clarify ("Table 1").

Reviewer #2: The first paragraph is headlined as describing sequence alignments, but it also includes structural data and discussions thereof. This is confusing.

Answer: Both sequence and domain architecture were used to rationalize specific recombinant constructs to produce, so we have renamed this section to avoid confusion as described above. We apologize for this.

Reviewer #2: Firstly, the discussion on the structure of the G5 domain may be better placed in the discussion (also in view of the fact that it was not solved here).

Answer: We have removed this discussion beyond citing our previous studies that have shown the distal G5 domain does not contribute to interactions with IgA1 previously explored for *S. pneumoniae* IgA1P.

Reviewer #2: Secondly, the discussion on the trypsin domain, based upon structure predictions and NMR studies appear out of place and better placed in a separate paragraph. This section also lacks any introductory information; NMR was performed on two constructs and I do not see an explanation why these two are chosen, nor do I see whether the NMR actually provides support for the theoretical models.

Answer: We apologize for lack of clarity, but it is in a separate paragraph within the first section of Results considering that it is the major difference between the *S. pneumoniae* IgA1P and the *G. haemolysans*. We produced multiple constructs comprising the trypsin-like domain both alone (GhTrp 684-896, GhTrp 651-896) and in the context of the M26 region (*G. haemolysans*). We hope that our inclusion of a table with these specific constructs clarifies this.

Reviewer #2. I think this data could be interesting, but then needs a better introduction in a separate paragraph. Finally, the proteins appear not active on a possible substrate (see next para), so is the protein purified, indeed, the correct conformation? And then based upon what assumptions? In the discussion (l. 337) it is mentioned that the trypsin domains was well-folded, but I miss a discussion on what assumption this claim is based.

Answer: The second paragraph within the section the first section of Results does begin with the major difference of these metallo-IgA1Ps, which we hope is now further clarified (“*G. haemolysans* metallo-IgA1P comparisons to other family members reveal specific constructs to interrogate”). We also apologize for taking the meaning of the observed NMR resonance dispersion for granted and have now included a brief sentence of what this means. Specifically, the dispersion of resonances immediately suggests that the protein is folded. Meaning, the fact that resonances have “dispersed” chemical environments strongly implies that the protein is folded, which is a general concept in macromolecular studies using NMR. Obviously, it is always possible that a domain is not correctly folded, but this is unlikely considering instabilities of misfolded proteins that often lead to precipitate or partial folding that would lead to relatively undispersed NMR spectra. If the reviewer believes that further evidence is necessary, we could also include analytical sizing that indicates a well-folded monomeric species as opposed to a misfolded protein that would migrate much larger. We have not included this considering that the NMR spectra illustrates an atomic-resolution perspective that indicates a well-folded protein.

Reviewer #2: The biochemical analysis of paragraph two is convincing in showing the catalytic activity of the purified domains. It lacks, however, a description why the E1848 was chosen for mutation into an inactive mutant. And I have to presume the same fragment of IgA1P was tested?

Answer: This is described now within the first paragraph of the first Results section (“*G. haemolysans* metallo-IgA1P comparisons to other family members reveal specific constructs to interrogate”). Specifically, within the first paragraph we describe the full M26 metallo-IgA1P (*G. haemolysans* IgA1P residues 907-2201) and the conserved catalytic residue (E1848) that was the basis for the catalytic mutant (*G. haemolysans* IgA1P E1848A). Within the second paragraph we describe the other three constructs that comprise the trypsin-like domain. All of these are listed in Table 1 to further provide clarity.

minor points:

Reviewer: In the abstract the authors claim stark differences, this seems to be overstated. The *Gemella* IgA1P is more open when no substrate is around, but shows a quite similar architecture to the streptococcus one when substrate is bound. This is corroborated by the very similar biochemical data that show pretty similar binding properties.

Answer: We completely agree and we have taken this over-interpretation out.

Reviewer #2: How widely the trypsin domain in Gemella IgA1P distributed among other species?

Answer: This is an interesting question, as this domain is widely distributed within other metallo-IgA1Ps in species that also include *Streptococcus pneumoniae* and *Streptococcus mitis*. While we had previously reported the only structurally solved ETA homologue from *S. aureus*, we have now included a statement that it is indeed found in other *Streptococcus* species within both Introduction and within the Discussion.

Reviewer: the authors mention in the introduction and discussion ZmpB-D as family members of the metallo-IgA1P proteases, but it remains unclear in what type of bacteria these are found, whether they exist besides the IgA1P and what the current study actually means for understanding these proteins.

Answer: In the Discussion, we have now elaborated on these other M26 protease family members. We have included the fact that they are broadly found in other species and specifically stated how their potential substrates may be identified based on our own studies of the metallo-IgA1Ps.

Reviewer: line 160; IgA1 substrate HC; should this not be IgA1P substrate HC?

Answer: We believe this is correct, as we are referring to the IgA1 substrate heavy chain (HC) that is cleaved by the IgA1P.

REVIEWERS' COMMENTS:

Reviewer #2 (Remarks to the Author):

Authors have done a good job revising the manuscript and all of my primary concerns on the technical side of manuscript have been addressed.

Reviewer #3 (Remarks to the Author):

I have re-reviewed the manuscript by Redzic et al. on the IgA protease of *Gemella haemolysans*. It provides a much improved presentation of their work, when compared to the previous version. I think it is now fully clear how their new structures support a structurally conserved gating mechanism to bind and cleave IgA molecules by these metalloproteases.

I have only a few comments to improve the manuscript:

1. P2, l51: typo (remove "}")
2. P5, l119: how can the sortase-induced cleavage of the motif place the protein "into the outer membrane" when we are discussing Gram-positive bacteria that do not have such an outer membrane? Needs re-phrasing.
3. P6, l155: replcae "produced" by "modelled". The structure was only predicted using modelling programs and then tested for giving a certain NMR spectrum. No structure was solved.
4. P8, l197-200: this is an awkward sentence probably resulting from some deleted text. I think it should be rephrased. ("ass" probably should be "assay"?)

One reviewer pointed out four corrections:

1. P2, l51: typo (remove "{")

Answer: We have removed this typo.

2. P5, l119: how can the sortase-induced cleavage of the motif place the protein "into the outer membrane" when we are discussing Gram-positive bacteria that do not have such an outer membrane? Needs re-phrasing.

Answer: This was meant to say "on the outside of the membrane" and we have changed this accordingly. Thank you.

3. P6, l155: replace "produced" by "modelled".

Answer: This has been changed.

4. P8, l197-200: this is an awkward sentence probably resulting from some deleted text. I think it should be rephrased. ("ass" probably should be "assay"?).

Answer: Correct and this has been changed.